Review Article

# The influence of AHR on immune and tissue biology

Brigitta Stockinger [1✉], Oscar E Diaz [1] & Emma Wincent [2]

## Abstract

**The aryl hydrocarbon receptor is a ligand dependent transcription factor which functions as an environmental sensor. Originally discovered as the sensor for man made pollutants such as 2,3,7,8-Tetrachlorodibenzo-p-dioxin (TCDD) it has recently gained prominence as an important mediator for environmental triggers via the diet or microbiota which influences many physiological functions in different cell types and tissues across the body. Notably AHR activity contributes to prevent excessive inflammation following tissue damage in barrier organs such as skin, lung or gut which has received wide attention in the past decade. In this review we will focus on emerging common AHR functions across cell types and tissues and discuss ongoing issues that confound the understanding of AHR physiology. Furthermore, we will discuss the need for deeper molecular understanding of the functional activity of AHR in different contexts with respect to development of potential therapeutic applications.**

**Keywords** Aryl Hydrocarbon Receptor; Cytochrome P4501; Tissue Repair; Intestinal Epithelium; Barrier Organs
**Subject Categories** Evolution & Ecology; Immunology

## Introduction

It has become increasingly clear in the past decades that environmental influences such as diet, lifestyle, or pollution have a substantial impact on physiological health and contribute, together with genetic factors, to shaping susceptibility to inflammatory diseases. One of the major molecular entry points for environmental factors is the aryl hydrocarbon receptor (AHR), a ligand dependent transcription factor belonging to the bHLH-PAS family of transcription factors which sense different aspects of the environment. Members of this family sense circadian rhythm, oxygen levels or external as well as endogenous planar aromatic hydrocarbons in the case of AHR. The AHR is an ancient protein with homologues in most major groups of modern bilaterian animals and about 600 million years of evolutionary development (Hahn et al, 2017). It was identified because of its role in the induction of drug-metabolising enzymes leading to the detoxification of xenobiotics such as polycyclic aromatic hydrocarbons (Poland et al, 1976). The major metabolising enzyme outside the liver, which is highly induced by AHR activation, is Cyp1a1, a

member of the cytochrome P4501 (Cyp1) family, often used as a biomarker for AHR activity. The toxicological aspects dominated the view on AHR over decades and restricted the interest to the fields of pharmacology and toxicology. This view has changed considerably with AHR now considered an important player in immunology as well as tissue and cancer biology.

With the discovery of 'physiological' AHR agonists derived from the diet or from tryptophan metabolism by some species of microbiota, the emphasis shifted to understanding the physiological functions of AHR in a wide range of cell types and tissues. AHR was thought to be ubiquitously expressed in all vertebrate cells, but more granular analysis on a per-cell basis rather than by measuring bulk tissue mRNA expression indicates that expression levels vary widely within tissues and conditions. For instance, AHR expression in adaptive immune cells of peripheral immune organs such as the spleen and lymph nodes is modest, with the highest expression in the Th17 CD4 T cell subset and negligible expression in most B cells, other T cell subsets and regulatory T cells (Treg) (Stockinger et al, 2014). In contrast, Treg in the gut express high levels of AHR (Ye et al, 2017). Myeloid cells such as macrophages, dendritic cells and eosinophils were reported to be highly positive for AHR, indicated by mRNA expression and functional studies (Goudot et al, 2017; Jin et al, 2014; Shinde et al, 2018; Wang et al, 2022). Furthermore, an AHR reporter mouse recently allowed direct comparisons on protein levels, showing exceptionally high AHR levels on myeloid cell types and other immune cells, particularly in the gut (Diny et al, 2022). In general, AHR expression is strong in barrier organs such as the lung, gut and skin, coinciding with the interface to ligand exposure. Conversely, it is low in peripheral lymphoid organs such as the thymus, spleen or lymph nodes, with the exception of innate immune cell types. Figure 1 gives an example of AHR distribution in different organs from an AHR-td-Tomato reporter mouse (unpublished data).

Little is known about the transcriptional control of AHR expression per se and most of the emphasis so far has been on AHR activation. However, it is clear that there are scenarios where AHR expression is reduced such as in inflamed gut tissue of patients with IBD (Monteleone et al, 2011). However, AHR expression per se may not be particularly meaningful unless it is below a threshold which precludes activation by ligands.

The identification of AHR-activating ligands is another complicated issue with many controversies in the literature. As there was until recently no crystal structure of the AHR ligand binding PAS-B domain, most determinations of ligands were conducted in vitro using highly sensitive luciferase constructs based on the consensus AHR response element (XRE). There are two problems with this approach. Firstly, the signal may be artificially amplified and not reflect activation under physiological conditions

[1]The Francis Crick Institute, London, United Kingdom. [2]Institute of Environmental Medicine, Karolinska Institute, Stockholm, Sweden. ✉E-mail: brigitta.stockinger@crick.ac.uk

**Glossary**

| | |
|---|---|
| ADME | description of the adsorption (A) of a chemical substance into the body, its distribution (D) and metabolism (M) within the body, and excretion (E) out of the body. |
| bHLH-PAS | A basic helix–loop–helix is a protein structural motif that characterises one of the largest families of dimerising transcription factors. A Per-Arnt-Sim domain acts as a molecular sensor and is found in a large number of organisms from bacteria to mammals. The name stems from the three proteins in which it was first discovered. |
| Bilaterian animals | animals (including humans) with two-sided symmetry and three body layers (endoderm, ectoderm and mesoderm). |
| Cryo-EM | Cryogenic electron microscopy, a microscopy technique applied to samples cooled to very low temperatures which preserves biological structures. |
| FICZ | 6-Formylindolo[3,2-b]carbazole, a high-affinity AHR ligand derived from tryptophan or tryptamine. |
| I3C | indole-3-carbinol. A chemical derived from cruciferous vegetables such as broccoli which undergoes a chemical transformation in the acidic environment of the stomach to produce high-affinity AHR ligands. |
| IEL | intraepithelial lymphocytes. A group of immune cells that are located in the intestinal epithelium, comprised of CD8+ lymphocytes as well as cells expressing the gamma/delta T cell receptor. |
| Receptor ligand | a compound, endogenously produced or xenobiotic, that exhibits specific binding to a receptor, leading to either activation of the receptor, i.e. agonistic response, or inhibition of receptor activity, i.e. antagonistic response. |
| TCDD | 2,3,7,8-Tetrachlorodibenzo-p-dioxin, a polychlorinated dibenzo-p-dioxin (sometimes shortened to dioxin) which is the prototypical xenobiotic ligand for AHR. |
| Xenobiotic | a chemical substance that is not naturally produced or expected to be present within an organism. It is often used in the context of man-made pollutants such as dioxins and polychlorinated biphenyls. |
| XRE | xenobiotic response element, a DNA sequence that contains the binding site for AHR. |

in vivo, and the in vitro approach is complicated by the fact that all tissue culture media contain tryptophan and, therefore may include its derivative 6-Formylindolo[3,2-b]carbazole (FICZ) (Rannug et al, 1987), a high-affinity AHR ligand which could be the reason for a positive signal (Rannug and Fritsche, 2006; Veldhoen et al, 2009). Secondly, it has been shown that many putative AHR ligands may not be ligands at all but rather inhibitors of the negative feedback of AHR activation via enzymes of the Cyp1 family. Cyp1 enzymes normally biotransform ligands to facilitate their excretion and thereby terminate AHR signalling (Wincent et al, 2012). Inhibition of their function thereby causes indirect AHR activation by endogenous ligands such as FICZ, whose biotransformation and elimination are inhibited due to the blockade of Cyp1 enzymatic activity (Schiering et al, 2018). That means that inferring the AHR agonist activity of chemicals in the absence of structural binding evidence must be interpreted with caution.

There have been many reviews of various aspects of AHR biology in the past 10 years. We will focus here on emerging common AHR functions across cell types and tissues and discuss ongoing issues that confound the understanding of AHR physiology. Furthermore, we will discuss the need for deeper molecular understanding of the functional activity of AHR in different contexts with respect to development of potential therapeutic applications.

# Main body

## Common AHR functions across tissues

AHR activation in the context of physiological ligands predominantly has anti-inflammatory functions due to its regulation of processes involved in promoting cell and tissue homoeostasis. This is not to say that AHR is never involved in inflammatory responses, which are reported, for instance, in the context of cancer or liver disease (Carambia and Schuran, 2021; Murray et al, 2014).

Physiologic ligands as opposed to man-made pollutants are thought to be biotransformed by AHR induced Cyp1 family members which catalyse their oxidative metabolism leading to their inactivation and excretion. This has been well demonstrated for the endogenous AHR ligand FICZ (Wincent et al, 2009), but has not been rigorously examined for a wider range of endogenous ligands.

Many man-made pollutants on the other hand are poor substrates for these enzymes and as a consequence circumvent the negative feedback of AHR activation by Cyp1 enzymes, interfering with its physiological function. Despite numerous reports of deleterious functions of environmental pollutants that activate AHR, the difference underlying the mode of action of physiological ligands vs pollutants remains mechanistically unexplained.

The anti-inflammatory consequences of AHR activation are well documented for the barrier organs skin, lung and gut by a multitude of publications and reviews, but detailed mechanistic insight is scarce. Many of the anti-inflammatory effects of AHR activation in the gut are attributed to the induction of IL-22 (Keir et al, 2020; Monteleone et al, 2011) in ILC or Th17 cells (Monteleone et al, 2012). There is now ample evidence for anti-inflammatory, tissue protective functions of AHR linked to multiple cell types and affecting diverse mediators. In some cases, AHR activity simply keeps cells alive, such as intraepithelial lymphocytes (IEL)(Li et al, 2011) or innate lymphoid cells type 3 (ILC3) (Qiu et al, 2012), albeit it is far from clear how this is achieved. In other cases, it endows particular mediators such as IL-22 production by Th17 cells (Veldhoen et al, 2008) or it influences the levels of effector proteins such as Granzyme B in CD8 tissue memory cells (Dean et al, 2023), or IEL (Maradana et al, 2023). AHR-deficient mice invariably show increased levels of tissue inflammation either directly mediated via inflammatory cytokines or indirect effects due to decreased barrier protection, e.g. by lack of IL-22, reduced levels of tight junction proteins, lack of cell populations such as IEL or ILC3 (Stockinger et al, 2021).

Tryptophan metabolites such as Indole-3-carboxyaldehyde (IAld) a precursor for the high-affinity AHR ligand FICZ (Smirnova et al, 2016) are generated by some microbiota strains and are implicated in gut mucosal integrity. Interestingly, IAld has recently been implicated in alleviating depressive symptoms in mice (Cheng et al, 2024), albeit

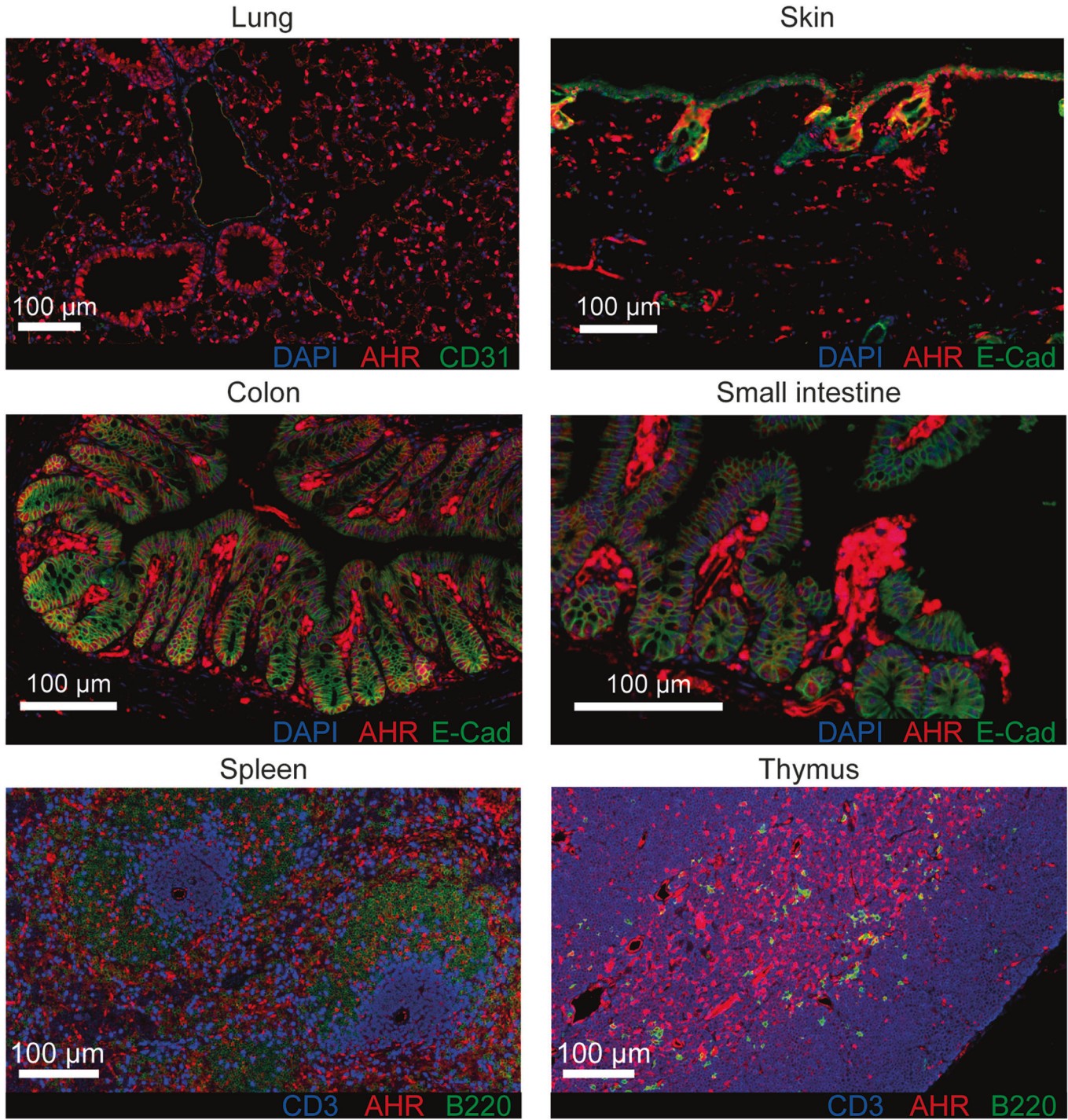

**Figure 1. Expression of AHR across murine tissues.**

Sections from different organs collected from a 12-week-old AHR reporter mouse (Diny et al, 2022) (AHR-td-Tomato) stained with different antibodies and DAPI as indicated in each panel.

it remained unclear whether the AHR pathway was involved. In mouse models of intestinal inflammation, some microbiota-derived tryptophan metabolites improve intestinal barrier function via modulation of host processes, such as actin regulatory proteins that control epithelial permeability (Scott et al, 2020). Microbiota-derived metabolites also alleviate liver inflammation through the IL-22 pathway (D'Onofrio et al, 2021), although mechanistic details are not defined.

AHR also plays a role in the autonomic nervous system of the gut, the enteric nervous system, where AHR expressed selectively

in colonic but not small intestinal neurons promotes gut peristalsis (Obata et al, 2020). AHR expression on colonic neurons is induced by microbiota and is absent in germfree mice. In contrast, intestinal epithelium expresses AHR also in germfree mice and is therefore not dependent on microbial induction of AHR expression. Neuron-specific AHR deletion or the absence of microbiota results in reduced peristaltic activity (Obata et al, 2020).

Intestinal T cells isolated from Crohn's disease patients have decreased levels of AHR, but upon exposure to AHR, ligands downregulate inflammatory cytokines and upregulate IL-22 (Monteleone et al, 2012). A recent study linking microbial metabolites implicated in the disease course of paediatric ulcerative colitis indicated that a decrease in tryptophan metabolites, many of which activate the AHR, is associated with moderate to severe ulcerative colitis (Schirmer et al, 2024). Other observations in pre-clinical and clinical settings of metabolic syndrome show an association with reduced generation of AHR-activating metabolites by the microbiota, which can be alleviated with AHR agonist-producing Lactobacilli strains (Natividad et al, 2018). While the general tendency in recent years has been an emphasis on the contribution of microbiota-derived metabolites on AHR function in the gut, it is important to also take into account dietary metabolites as well as host-derived metabolites (Hubbard et al, 2015). The expression of Cyp1a1 decreases from the duodenum to the colon, whereas the microbial density in the colon far exceeds that of the small intestine (Zhou et al, 2023). This suggests that dietary-derived AHR ligands may be the prevalent source of AHR activation in the small intestine, whereas microbial metabolites may play a more prominent role in the colon. However, the composition of the microbiota and, thereby, its potential to generate AHR-activating metabolites is certainly influenced by diet.

Infection with *Cryptosporidium* which causes severe diarrhoea, especially in children and immunocompromised or malnourished individuals, is influenced by indole levels in the gut. Volunteer studies established an inverse correlation between susceptibility to Cryptosporidium infection and faecal indole levels prior to infection (Chappell et al, 2016). AHR-deficient mice, as well as mice fed purified diet deprived of AHR ligands, were highly susceptible to Cryptosporidium infection due to the reduction in IELs which depend on AHR signalling for survival and are essential to control the infection. Supplementation of diet with the pro-ligand indole-3-carbinol (I3C), a dietary component from crucifer-ous vegetables, restored IEL levels and made newborn mice resistant to infection (Maradana et al, 2023).

Another site with very high AHR expression and constitutive activation of Cyp1a1 are endothelial cells. Vascular abnormalities and effects on cardiovascular physiology are a prominent feature of AHR-deficient mice (Lahvis et al, 2000; Zhang, 2011), but in adult mice, endothelial cells are normally quiescent, although the high expression of Cyp1a1 indicates constant exposure to AHR activation which may be facilitated by shear stress (Han et al, 2008; McMillan and Bradfield, 2007). Recent studies established that AHR signalling in endothelial cells promotes cellular quiescence and protects against cellular leakage, whereas disruption of AHR signalling causes a dysregulated stress response with inflammation (Major et al, 2023; Wiggins et al, 2023). Interestingly, activation of AHR by different xenobiotics (Kopf and Walker, 2009) is also associated with cardiovascular adversities.

While this is not observed with 'natural' ligands such as FICZ, blocking the Cyp1-dependent metabolism of FICZ, which prolongs AHR activation, causes a similar phenotype (Wincent et al, 2016). Although the mechanisms underlying these effects are not fully understood, a study using zebrafish demonstrated that constitutive AHR activation in cardiomyocytes recapitulates not only the developmental cardiotoxicity observed with the dioxin TCDD, but also many other endpoints of toxicity produced by TCDD in zebrafish larvae (Lanham et al, 2014). Hence, activating and depleting AHR function both affect the cardiovascular system, but likely through different downstream events.

Activation of the AHR pathway is also important in maintaining skin barrier homoeostasis. Dysregulation of AHR by either genetic deficiency or by excess activity of the downstream negative regulator Cyp1a1 causes skin pathology, and patients suffering from psoriasis display reduced activity of the AHR pathway and increased enzymatic activity of Cyp1a1 compared with healthy donors (Kyoreva et al, 2021). In the skin, commensal bacteria-derived metabolites activate AHR in keratinocytes and contribute to repair the skin barrier in disease models (Uberoi et al, 2021), but the skin is also a site where the tryptophan derivative FICZ is generated as a tryptophan photo-product, as well as through light-independent pathways (Rannug and Fritsche, 2006; Smirnova et al, 2016).

A consistent theme for the functional activity of AHR is its involvement in regenerative processes. AHR activation promotes differentiation of injured intestinal epithelium (Shah et al, 2022) or lung epithelium (Morales-Hernandez et al, 2017), and inhibition of AHR in haematopoietic stem cells expands CD34+ cells with the ability to engraft immunodeficient mice (Boitano et al, 2010) at the expense of differentiation.

While AHR deficiency accelerates stem cell proliferation both in the lung (Morales-Hernandez et al, 2017) and in the intestinal epithelium, the failure to reconstitute a functioning epithelium following injury could be ascribed to a defect in efficient differentiation in the absence of AHR (Shah et al, 2022). The combined effect of stem cell over-proliferation with chronic inflammation caused by a defective barrier makes AHR-deficient mice susceptible to malignant transformation (Metidji et al, 2018).

## Physiological AHR functions and underlying mechanisms

Given the widespread expression of AHR and its range of functions in different tissues and cell types, it seems surprising that the full-body knockout of AHR does not compromise the survival of the mice. AHR is an ancestral regulator of developmental processes such as neural development, which is evolutionarily conserved in invertebrates (Burgess and Duncan, 1990; Huang et al, 2004; Kim et al, 2006). A prominent feature of AHR deficiency in mice are defects in vascular structure, the liver and the heart (Harstad et al, 2006; Lahvis et al, 2000; Vasquez et al, 2003). While in the gut, there are no obvious macroscopic changes visible in AHR-deficient mice, they do not survive well in a conventional animal facility (Fernandez-Salguero et al, 1997). Development of organs such as the intestine is proceeding normally in the absence of AHR, but AHR functions become critical in the adult stages for restoration of homoeostasis after injury. Injury repair mechanisms, in many ways, recapitulate processes operative in development. Thus, following damage of intestinal epithelial cells, committed epithelial cells dedifferentiate and reprogram back into a foetal-like stem cell

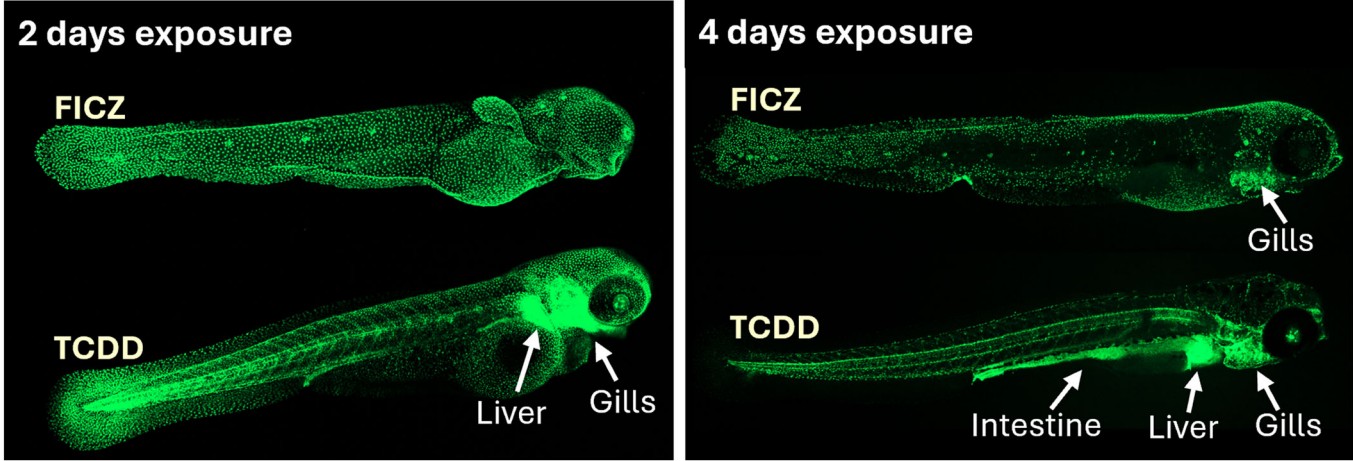

**Figure 2. Distribution of AHR2 activation in zebrafish embryos after exposure to FICZ and TCDD.**

Images of a Cyp1a reporter strain (Tg(Cyp1a:EGFP)) exposed to FICZ (50 nM) and TCDD (0.01 nM), respectively, from 1 day post fertilisation through the embryo water. At 2 and 4 days of exposure, images were taken by confocal microscopy. In the case of FICZ, AHR2 activation was restricted to the outer barrier on day 2, of which very little was remaining on day 4. In contrast, TCDD caused an activation mainly in internal organs already at day 2, especially in the liver and gills, which was still strong at day 4, when also the intestinal tract was highly activated.

state (Jadhav et al, 2017). In the absence of AHR, the subsequent differentiation of stem cells is compromised (Shah et al, 2022).

While early developmental processes are less dependent on AHR, in an adult tissue where repair is time-critical due to a multitude of exacerbating factors AHR becomes essential. For instance, the rapid regeneration of a defective gut or skin barrier is time-sensitive, or it will be breached by the microbiota with serious consequences such as inflammation. This emphasises the time- and context-dependent impact of activation and depletion of AHR. Whereas depletion of AHR during development does not result in lethality or overt severe effects, prolonged activation of AHR does, as exemplified by the developmental toxicity observed upon exposure to persistent xenobiotic ligands such as TCDD and PCB126 (King-Heiden et al, 2012). Xenobiotic activation of the AHR at adult life stages causes less detrimental effects compared with the developmental stages, but the impact on restoration of homoeostasis after injury has not been investigated.

In epithelial cells, AHR activity curtails the proliferation of stem cells and promotes differentiation (Shah et al, 2022; Zhou et al, 2023), in endothelial cells, it promotes endothelial cell differentiation and vascular repair (Major et al, 2023; Wiggins et al, 2023). There are numerous targets downstream of AHR that are invoked in functional outcomes of AHR activation, but the precise molecular mechanisms underlying these remain largely undefined, and we currently do not have a coherent picture of which molecular mechanisms are direct or indirect consequences of AHR activation. The initial simplified picture of AHR binding at conserved xenobiotic response elements (XRE), which are frequent in the genome, is clearly not sufficient to explain the actions of AHR. In many studies, including our own, representation of AHR binding sites in the genome which contain an XRE site are a fraction of all binding sites identified in chromatin immunoprecipitation (ChIP) sequencing studies (Dere et al, 2011; Shah et al, 2022). Most of the early studies were done in cell lines activated with the prototypical AHR ligand TCDD, but there is a notion now

that takes into account potential differences in the effects of AHR activation by xenobiotic or endogenous ligands based on their tissue distribution and metabolism which is rapid for endogenous ligands and very slow for many xenobiotic ligands such as TCDD (Boule et al, 2018). An example of this is given in Fig. 2 which shows the distribution of Cyp1a expression in zebrafish as a consequence of AHR activation by either TCDD or FICZ.

In our analysis of AHR CHIP peaks in mouse intestinal organoids (Shah et al, 2022), we identified only about 10% with the classical XRE motif. 80% of the peaks were located in introns or intergenic regions in accordance with previous data (Dere et al, 2011), suggesting alternative mechanisms of AHR-mediated gene regulation, possibly through tethering mechanisms involving other transcription factors or by DNA looping. We showed that AHR is required to restrict chromatin accessibility to factors associated with regeneration (e.g. *Yap/Tead*) and could function as a transcriptional activator and a transcriptional repressor for pro-differentiation factors (*Cdx2*) and proto-oncogenes (*Sox9* and *Myc*), respectively. This is highly dependent on the context and cell type, which adds up to a bewildering diversity of functional impacts across the body. The interaction of the AHR/ligand complex with tissue and cell-specific sets of nuclear cofactors influences the agonist or antagonist activity of the ligand, which are not readily predictable (Safe et al, 2020). A further complication is that AHR may have non-genomic functions (reviewed in (Bock, 2020). However, examples of these are restricted to cells in vitro and have not been substantiated in vivo. A genetically modified mouse model in which AHR is present in the cytoplasm, capable of binding ligand but not moving to the nucleus and not binding DNA (Wilson and Bradfield, 2021) would seem ideal to resolve these issues, but so far, no data on alternative AHR signalling in this mouse model have been published.

Altogether it is difficult at present to claim that there is a full understanding of the mechanisms underlying AHR function.

## AHR as therapeutic target

With the realisation of the wide range of physiological functions of AHR came an increasing interest in targeting this pathway for inflammatory diseases (Accioli et al, 2023; Chen et al, 2023). One of the areas of interest is targeting AHR in cancer (Paris et al, 2021), albeit the role of AHR in cancers is ambiguous as there is evidence for a tumour-suppressive role as well as potential involvement in tumour progression (Murray et al, 2014).

An obvious choice of target is the skin, which is affected by environmental pollutants linked to inflammatory diseases such as atopic dermatitis, pigmentation disorders or acne (reviewed in (Accioli et al, 2023)) albeit the mechanistic details of AHR involvement remain unclear. There is evidence that skin exposure to 'natural' compounds such as polyphenols and flavonoids may control proinflammatory skin reactions and ameliorate conditions such as psoriasis (Di Meglio et al, 2014). Tapinarof, a stilbene, is the first presumed AHR ligand in clinical use for the treatment of psoriasis and atopic dermatitis (Bobonich et al, 2023), albeit its structure makes it unlikely to be a direct AHR ligand. Potentially, it might function as an indirect AHR activator via the previously mentioned inhibition of Cyp1a1-mediated metabolism of naturally present ligands such as FICZ or skin microbiota metabolites (Wincent et al, 2012).

The wide range of postulated AHR ligands gave rise to the concept of selective AHR modulators with agonist or antagonist activities depending on the cell/tissue or organ context (Safe et al, 2020). This leaves a wide-open field for potential ligand-focused therapies. A rational design approach involving computational modelling and lead optimisation gave rise to a putative agonist with favourable pharmacological properties that inhibited DSS induced colitis in mice (Chen et al, 2020). There are also many studies on the cancer-preventative activities of the dietary pro-ligand I3C and its bioactive derivative 3.3'diindolylmethane (DIM) (reviewed in (Reyes-Hernandez et al, 2023)). The other I3C derivative indolo[3,2-b]carbazole (ICZ) is generated in lower amounts but is a high-affinity AHR ligand that in contrast to DIM is rapidly metabolised (Nguyen and Bradfield, 2008; Schiering et al, 2017). On the other hand, there are several approaches underway to inhibit AHR activity in tumour settings, with antagonists targeting the putative suppressive effect of AHR on anti-tumour immunity and resistance to immune checkpoint inhibitors (Campesato et al, 2020; Kober et al, 2023; McGovern et al, 2022). It is difficult to rationalise the underlying rationale for these divergent viewpoints on AHR action in a tumour setting, but it will be important to carefully characterise the functional effect of AHR activation or inhibition in different cell types present in the tumour environment.

Ongoing difficulties in moving forward with AHR-focused therapeutic interventions are the lack of a clear understanding of what true AHR agonists are as well as our lack of detailed molecular understanding of the toxicity associated with AHR-activating xenobiotics.

Until recently, there were substantial difficulties in solving the structure of the ligand binding AHR-PAS-B region since this region could not be expressed in a soluble form, but in 2022 a Cryo-EM study showed the long-awaited structure of the human AHR-PAS-B domain with the ligand indirubin (Gruszczyk et al, 2022). These data provided evidence of a primary binding site for AHR ligands but also pointed to a secondary binding site which potentially could

accommodate larger compounds. It will be interesting to investigate if additional ligands can bind to the secondary site and what impact they will have on AHR signalling.

How much activation is beneficial versus detrimental is another key question that needs to be resolved for the therapeutic potential of AHR as well as for assessing the risks of exposure to environmental pollutants acting through this receptor. Previously, the toxicity of xenobiotic AHR activators have been attributed to their persistence in the body, and consequently, the prolonged duration of AHR activation. However, we do not know yet if the effects of exogenous AHR ligands are solely due to the duration of AHR signalling. This needs to be explored using experimental models evaluating critical physiological functions we now know AHR to have. An additional point to consider is the distribution of the agonist in different tissues. As illustrated in Fig. 2, AHR agonists such as FICZ and TCDD that are of similar potency cause completely different spatial distribution of activation due to their distinct ADME properties (unpublished data), highlighting the importance of fully understanding the target organ distribution of ligands, irrespective of the mode of application, when studying the role of AHR in health and disease.

## Pending issues

New discoveries on different aspects of AHR biology keep accumulating, and there is now a strong interest in targeting the AHR pathway in different diseases. Nevertheless, there are still fundamental issues that need to be resolved before many of these may become realistic options. In Box 1 we have summarised some pending issues in the field that await resolutions. This ranges from a detailed understanding of mechanisms underlying AHR action determined in primary cells rather than cell lines, to more granularity in understanding AHR functions in tissue repair and the evaluation of the relative impact of xenobiotic vs natural AHR ligands in such processes, the distribution of ligands in target tissue and their clearance through the Cyp1/AHR feedback mechanisms. Now that the crystal structure of a ligand bound to the AHR-PAS-B domain has been elucidated it will be important to characterise further ligands. In addition, exploration of the reported second binding site and an assessment of the functional impact of potential binding of additional ligands will likely be forthcoming.

## Peer review information

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

coordinates mouse small intestinal epithelial cell programming. Lab Invest
103:100012

## Acknowledgements

This work was supported by the Francis Crick Institute, which receives its core
funding from Cancer Research UK (CC2016), the UK Medical Research Council
(CC2016), and the Wellcome Trust (CC2016) and a Wellcome Investigator
Award to BS (210556/Z/18/Z). EW is funded by the Swedish Research Council
VR (2020-03418) and Svenska Forskningsrådet Formas (Formas FR-2021/0005).

## Author contributions

**Brigitta Stockinger**: Conceptualisation; Data curation; Formal analysis;
Supervision; Funding acquisition; Validation; Writing—original draft; Project
administration; Writing—review and editing. **Oscar E Diaz**: Data curation.
**Emma Wincent**: Conceptualisation; Data curation; Writing—review and editing.

## Funding

## Disclosure and competing interests statement

The authors declare no competing interests.

