## [Peer Review File · EMBO Molecular Medicine]

The influence of AHR on immune and tissue biology

Brigitta Stockinger, Oscar Diaz, and Emma Wincent

Corresponding authors: Brigitta Stockinger (Brigitta.Stockinger@crick.ac.uk) , Emma Wincent (emma.wincent@ki.se)

Review Timeline:

Submission Date:	3rd May 24
Editorial Decision:	10th Jun 24
Revision Received:	15th Jul 24
Editorial Decision:	14th Aug 24
Revision Received:	16th Aug 24
Accepted:	19th Aug 24

Editor: Zeljko Durdevic

Transaction Report:

10th Jun 2024

Dear Dr. Stockinger,

Thank you for the submission of your manuscript to EMBO Molecular Medicine. We have now received feedback from the two reviewers who agreed to evaluate your manuscript. As you will see from the reports below, the referees are positive about its interest and timeliness, however, they also raise serious criticisms that should be addressed in a revised manuscript. Further consideration of a revision that addresses reviewers' concerns in full will entail a second round of review. Particular focus should be given to:

- 1) Providing more balanced view on AHR signaling and mode of action.
- 2) Including all relevant citations and citing primary literature.
- 3) Restructuring the manuscript to avoid redundancy.
- 4) Tightening of the section "AhR as a therapeutic target".

I would also like to ask you to add the following items to your revised article:

- 1) Glossary: As for other terms please explain following ones:

- I3C: indole-3-carbinol
- IEL: intraepithelial lymphocytes
- XRE: xenobiotic response element

- 2) Figure 3 should be changed to Box 1 and added to the main manuscript file.

- 3) Please indicate in the figure legends the source of the images - your own original staining or adopted from previous publications. If the images originate from previous publication(s) the article(s) should be cited in the figure legends too.

- 4) Please provide Disclosure and competing interests statement. We updated our journal's competing interests policy in January 2022 and request authors to consider both actual and perceived competing interests. Please review the policy <https://www.embopress.org/competing-interests> and update your competing interests if necessary.

I hope that the referees' comments do not prove too problematic to address and I look forward to reading your next version.

Zeljko Durdevic

*** IMPORTANT INFORMATION ***

- 1) a .doc formatted version of the manuscript text (including Figure legends and tables)
- 2) Separate figure files
- 3) a letter INCLUDING the reviewer's reports and your detailed responses to their comments.

Also, and to save some time should your paper be accepted, please read below for additional information regarding some

features of our research articles:

1) Glossary: EMBO Molecular Medicine articles will be accompanied by a glossary explaining some of the terms used for laymen. I identified the following:

_____, _____, _____

Could you please help us in identifying terms that may need an "explanation" other terms that we can add to the glossary.

2) For more information: This is a short list of related web links for further consultation by the readers. Could you identify some relevant ones? Examples are patient associations, OMIM related links, databases, authors websites, etc.

3) Pending issues: At the end of each article we will have a box highlighting issues that still need further studies and where research efforts should converge (we call this the Pending issues box). From my reading I would say:

but I can see there may be many more. Could you work on this as well?

4) Disclosure and competing interest statement: Please include a statement declaring any competing commercial interests in relation to your submitted work.

5) Please note that we now mandate that all corresponding authors list an ORCID digital identifier. This takes <90 seconds to complete. We encourage all authors to supply an ORCID identifier, which will be linked to their name for unambiguous name identification.

Currently, our records indicate that the ORCID for your account is 0000-0001-8781-336X.

Link Not Available

-

Thank you,

Zeljko Durdevic

***** Reviewer's comments *****

Referee #1 (Remarks for Author):

This is a thoughtful and well written narrative review. The AHR is topic that has been the subject of many review articles in the past several years, and the authors gave a fresh spin to their article. I particularly appreciate that the authors highlight some of the current challenges and opportunities in the field, including some possibly 'inconvenient truths,' such as the limitations of relying on in vitro screening assays to identify putative AHR ligands. While the review is highly readable and compelling, there a few aspects that could be even clearer and a few things that need to be modified to further improve the review.

The main criticism is that the generalizability of the anti-inflammatory consequences of AHR activation is overstated. The authors ignore papers and data that report the opposite, and they also conflate immune suppression (e.g., dampened CD8 T cell response) with anti-inflammatory effects.

Figures 1 and 2 should probably be removed. Figure 1 is not mentioned in the text of the article, and Figure 2 does not seem necessary. (if the authors disagree and retain, then methodological details including source animals along with citations need to be added to the figure legends).

There are some statements that would be more robust with citations. For example, the very strong statement "Physiologic

ligands as opposed to man-made pollutants are readily bio-transformed by AHR induced Cyp1 family members which catalyse their oxidative metabolism leading to their inactivation and excretion" requires citations. It would also benefit from the caveat that for many "physiological AHR ligands" their metabolism (or at least their AHR-driven metabolism) has not been rigorously examined, particularly in vivo. Thus, this statement by the authors is not strongly supported by existing data. That said, I agree with their point and believe it is an important issue for the field.

The article as a whole could cite publications of others a bit more generously. There are of instances in which, rather than citing the first (or at least some of the first) reports of an observation, the authors instead cite their own more recent research. One example is Diny et al 2022 for citing that AHR expression in DCs, macrophages and other cell types in gut and lung. Several other prior publications reported these findings. As another example, when pointing out that FICZ's in vivo metabolism can be slowed by blocking or eliminating Cyp1a1 expression was previously reported by others, yet the authors only cited themselves.

Minor points to consider:

The authors raise a very central point in that we don't understand how much AHR activation is beneficial versus detrimental. However, the statement "Previously, the toxicity of xenobiotic AHR activators have been attributed to their persistence in the body, and consequently prolonged duration of AHR activation. This association has however not been clearly confirmed using experimental models evaluating the critical functions we now know AHR to have" may be misunderstood. For instance, this could imply that there is not much research on the mechanisms via which some exogenously derived AHR ligands cause pathophysiological consequences. I think they may be trying to make a more nuanced point: we don't yet know if the effects of exogenous chemicals that bind AHR is due solely to the duration of AHR signaling. I agree that this is the case. Yet, there are studies that show for some PAHs (e.g., benzo(a)pyrene), it is their metabolites that cause toxicity, not the parent compound. Mentioning this bolsters their overarching point about the complexity of AHR signaling, and the many unknowns that have yet to be resolved.

2,3,7,8-tetrachlorodibenzo-p-dioxin is misspelled (it is missing 'para' (-p-) between dibenzo and dioxin).

Referee #2 (Remarks for Author):

The topic of the review article entitled "The influence of environmental signals via AHR on immune and tissue biology" is timely, very interesting and potentially important, in particular with regards to the potential targeting of AHR in clinical situations. However, this article is not well structured and partially redundant, especially with regards to the authors' hypothesis that the majority of "so-called" AHR ligands act indirectly via inhibition of CYP1A1 and accumulation of endogenous AHR ligands. In my eyes, the authors do not pay sufficient attention to the growing list of studies investigating the different facets of AHR signaling as well as the AHR-dependent mode-of-action of certain chemicals. In its current form, the article seems to reflect the very subjective opinion of the authors, while lacking an open and critical discussion of findings that do not necessarily align with this opinion.

Points of concern:

1. Manuscript title is misleading. This article pays little to no attention to the impact of environmental signals on AHR.
2. Page 2, 2nd paragraph, l. 12: "The major metabolising enzyme which is highly induced by AHR activation is Cyp1a1,..."

This is probably true for some but not all tissues (liver > CYP1A2), cell-types, and conditions.

3. Page 2, 2nd paragraph, l. 14: "The detoxification aspect dominated the view on AHR over decades and restricted the interest to the fields of pharmacology and toxicology. This view has changed considerably over the last decade..."

Not true. Detoxification is not the same as toxicity. In general, this statement omits numerous important discoveries in the AHR field in various contexts, including immunology and cancer biology, that have been published way before 2014.

4. Page 3, l. 11 from top: "..., most determinations of ligands were conducted in vitro using highly sensitive luciferase constructs based on the consensus AHR response element (XRE). There are two problems with this approach. Firstly, the signal may be artificially amplified and not reflect activation under physiological conditions in vivo, and the in vitro approach is complicated by the fact that all tissue culture media contain tryptophan and therefore may include its derivative 6-formylindolo[3,2-b]carbazole (FICZ)(Rannug et al, 1987), a high affinity AHR ligand which could be the reason for a positive signal(Rannug & Fritsche, 2006; Veldhoen et al, 2009). Secondly, it has been shown that many putative AHR ligands are not ligands at all but rather inhibit the negative feedback of AHR activation via enzymes of the Cyp1 family. Cyp1 enzymes normally biotransform ligands to facilitate their excretion and thereby terminate AHR signaling(Wincent et al, 2012)."

I strongly recommend to attenuate these statements. For sure there is a long list of compounds that are called AHR ligands based on respective reporter gene data only. However, on the other hand one could easily list several dozens of AHR ligands that have been identified by means of radioactive ligand binding assays. Is this also true for CYP1A1 substrates as indicated by "many putative AHR ligands are not ligands..."? And if so, I assume it has been shown in vivo rather than in an artificial in vitro assay? Also, I think that the maximum CYP1A1 response that can be expected by accumulating FICZ is rather low in comparison to the dose-dependent induction by numerous of the "so-called" AHR ligands. In addition, I want to emphasize that the authors' idea that more or less all AHR ligands are metabolized by CYP1 enzymes is simply not true and I'm not only thinking of metabolically stable dioxins. The phase I reaction is thought to polarize lipophilic chemicals, accordingly, it is unlikely that CYP1 enzymes metabolize polar "AHR ligands", such as tryptophan metabolites (doi:10.1177/1178646923118250).

5. Page 3, l.10: "Physiologic ligands as opposed to man-made pollutants are readily biotransformed by AHR induced Cyp1 family members which catalyse their oxidative metabolism leading to their inactivation and excretion. In contrast, man-made pollutants are largely poor substrates for these enzymes and as a consequence circumvent the negative feedback of AHR activation by Cyp1, interfering with its physiological function."

This is again a very general statement. Please provide references for the CYP1A1 catalyzed metabolism of physiological ligands other than FICZ/ICZ. Also, some PAHs as well as man-made drugs are quite good substrates for CYP1 enzymes.

6. Page 3, l.6 from bottom: "Despite numerous reports of deleterious functions of environmental pollutants that activate AHR, the difference underlying the mode of action of physiological ligands vs pollutants remains mechanistically unexplained, but it is likely related to dysregulated kinetics of AHR signaling."

This is again a very general statement and partially not true. What about the established Adverse Outcome Pathways involving AHR activation by environmental pollutants (see websites of OECD, US EPA, etc.)? In addition, the mechanisms by which polyaromatic hydrocarbons are converted into highly genotoxic and/or redox-cycling metabolites are well-known. In addition, the authors should explain what they mean when stating "dysregulated kinetics of AHR signaling". In the context of xenobiotic metabolism and the definition of toxicokinetics and -dynamics, this is somehow confusing.

7. Page 3, l. 2 from bottom: "The anti-inflammatory consequences of AHR activation are well documented..."

Please specify. The TCDD-induced and AHR-mediated upregulation of COX2, for instance, is probably not an anti-inflammatory event. I assume this statement refers to physiological AHR signaling?

8. Pages 3-5: A few comments to the passage "Common AHR functions across tissues":

In the context of gut-derived microbial metabolites, it might be fair to mention that a lot of these compounds do not exclusively act via AHR but also via PXR (doi:10.1016/j.immuni.2014.06.014).

When discussing the role of AHR in endothelial cells and mentioning the vascular effects observed in AHR KO mice, is it of any relevance that those vascular phenotypes are not present in the AHR KO rat and thus might be species-specific? (doi:10.1016/j.taap.2013.06.024).

9. Pages 6-7: The chapter "Mechanisms of AHR function" appears rather premature and does not deliver what it promises. In fact, it is more or less a continuation of the previous chapter, enumerating phenotypes of AHR knockout etc. A thorough and comprehensive overview about the manifold mechanisms of AHR signaling and associated functions is missing. The authors did not spend a single word on all the alternative pathways of AHR that have been identified in numerous studies to influence the outcome of AHR activation during immune and developmental processes as well as inflammatory responses. The authors instead focus on potential XRE and ADME parameters of the ligands, which for sure helps to explain some but certainly not all of the dichotomous effects evoked by AHR.

10. Page 6, l. 15 from bottom: "Xenobiotic activation of the AHR at adult life stages causes less detrimental effects compared with the developmental stages..."

Somehow weird to weigh detrimental health effects, especially when thinking in the human context. What about the central role of AHR in PAH exposure-associated cancers and other detrimental health consequences? Or do the authors only consider fish and mouse models when stating that?

11. Page 8, l. 110: "... it is conceivable that the therapeutic action of tapinarof is not due to direct AHR activation but rather inhibition of Cyp1a1 mediated metabolism of naturally present ligands such as FICZ or skin microbiota metabolites."

This is highly speculative and should be rephrased. Do the authors think that coal tar acts in a similar manner?

Minor corrections/comments:

12. Abstract, l. 3: Please write "2,3,7,8-Tetrachlorodibenzo-p-dioxin" instead of "2,3,7,8-Tetrachlorodibenzodioxin"

13. Abstract, 3rd sentence: One or more words are missing "Notably AHR activity is important in preventing excessive inflammation following tissue damage in barrier organs such as skin, lung or gut has received wide attention in the past decade."

14. Page 2, 2nd paragraph, l. 10: "It was identified because of its role in the induction of drug metabolizing enzymes leading to the detoxification of xenobiotics such as polycyclic aromatic hydrocarbons (Poland et al, 1976)."

Wrong reference, please correct. Poland et al. have discovered AHR as a protein that binds to TCDD, which is neither a PAH nor efficiently detoxified via AHR/CYP1.

Referee #3 (Remarks for Author):

This is an important review topic. Clarification of AHR function with a modern understanding of physiological vs. the traditional understanding derived from toxicological studies is needed. The story is complex, and thus requires clarity in writing. The ideas expressed in the review are important and timely. The writing can be improved. See specifics below:

Introduction:

"major molecular entry points for environmental factors is the aryl hydrocarbon receptor (AHR), a ligand dependent transcription factor belonging to the bHLH-PAS family of transcription factors which sense different aspects of the environment."

Does this mean that the entire family senses the environment, or is this a specific function of the AHR. Please clarify.

Main Body

AHR expression on colonic neurons requires an intact microbiota which contrasts with intestinal epithelium which expresses AHR also in germfree mice.

I am not sure what this means. Please clarify

Mechanisms of AHR Function

I am not sure the first couple of paragraphs of this section should be characterized as mechanisms. For me mechanism is more molecular-based, biochemical and molecular processes. This seems more overall inflammatory systems.

Fig.2 which shows the distribution of Cyp1a expression in zebrafish as a consequence of AHR activation by either TCDD or FICZ.

This is an essential piece of the authors' argument. A more detailed explanation of the consequences of this kind of function is warranted.

Altogether it is difficult at present

to claim that there is a coherent picture of how AHR activation plays out in molecular terms. I am not sure what this actually means.

AhR as a therapeutic target.

This is an important part of the review. However, the section seems to be a little chaotic and wandering. It needs better organization to understand the potential roles for AhR-based therapeutics across different disease states and cell types.

Point-by-Point Response

We would like to thank all reviewers for their constructive comments and suggestions on our manuscript. We hope that our revised manuscript and the point-by-point response to the comments will alleviate their concerns. Changes in the manuscript have been marked in yellow to make it easier to go through the revision.

Referee #1 (Remarks for Author):

This is a thoughtful and well written narrative review. The AHR is topic that has been the subject of many review articles in the past several years, and **the authors gave a fresh spin to their article. I particularly appreciate that the authors highlight some of the current challenges and opportunities in the field, including some possibly 'inconvenient truths,' such as the limitations of relying on in vitro screening assays to identify putative AHR ligands.** While the review is highly readable and compelling, there a few aspects that could be even clearer and a few things that need to be modified to further improve the review.

We thank the Reviewer for the positive statements and endeavour to answer the points of criticism below.

The main criticism is that the generalizability of the anti-inflammatory consequences of AHR activation is overstated. The authors ignore papers and data that report the opposite, and they also conflate immune suppression (e.g., dampened CD8 T cell response) with anti-inflammatory effects.

We have toned down our statement and included a sentence mentioning that AHR may be involved in inflammatory responses too. Our reason to 'conflate' dampened CD8 responses with anti-inflammatory effects stems from the findings that excessive CD8 T cell responses are often contributing to tissue damage if not regulated properly.

Figures 1 and 2 should probably be removed. Figure 1 is not mentioned in the text of the article, and Figure 2 does not seem necessary. (if the authors disagree and retain, then methodological details including source animals along with citations need to be added to the figure legends).

We would like to keep these figures as they both illustrate different important aspects of AHR. They are mentioned in the text but we have made this clearer and also indicated that these are currently unpublished data from both our labs. The figure legends have also been extended to provide more information.

There are some statements that would be more robust with citations. For example, the very strong

statement "Physiologic ligands as opposed to man-made pollutants are readily bio-transformed by AHR induced Cyp1 family members which catalyse their oxidative metabolism leading to their inactivation and excretion" requires citations. It would also benefit from the caveat that for many "physiological AHR ligands" their metabolism (or at least their AHR-driven metabolism) has not been rigorously examined, particularly in vivo. Thus, this statement by the authors is not strongly supported by existing data. That said, I agree with their point and believe it is an important issue for the field.

We have to concede that direct ligand metabolism by Cyp1 enzymes has so far only been shown for FICZ, and to some extent ICZ, so we have toned down this sentence. Nevertheless, the clear in vivo demonstration that overexpression of Cyp1a1 results in loss of ligands and phenocopy of an AHR null serves as proof that more of the endogenous ligands in the body must be degraded by excessive Cyp1a1 activity.

The article as a whole could cite publications of others a bit more generously. There are of instances in which, rather than citing the first (or at least some of the first) reports of an observation, the authors instead cite their own more recent research. One example is Diny et al 2022 for citing that AHR expression in DCs, macrophages and other cell types in gut and lung. Several other prior publications reported these findings. As another example, when pointing out that FICZ's in vivo metabolism can be slowed by blocking or eliminating Cyp1a1 expression was previously reported by others, yet the authors only cited themselves.

We apologize if we appear too self serving in our citations. What we really meant was that the Diny study is really the first and so far only one in which a direct comparison of AHR protein levels was performed. We have now included citations of previous studies that mention high AHR expression in myeloid cells on RNA level and deduced by functional studies. With respect to slowing of FICZ metabolism by eliminating Cyp1 enzymes, however, we are not aware of any other published studies in vivo showing this.

Minor points to consider:

The authors raise a very central point in that we don't understand how much AHR activation is beneficial versus detrimental. However, the statement "Previously, the toxicity of xenobiotic AHR activators have been attributed to their persistence in the body, and consequently prolonged duration of AHR activation. This association has however not been clearly confirmed using experimental models evaluating the critical functions we now know AHR to have" may be misunderstood. For instance, this could imply that there is not much research on the mechanisms via which some exogenously derived AHR ligands cause pathophysiological consequences. I think they may be trying to make a more nuanced point: we don't yet know if the effects of exogenous chemicals that bind AHR is due solely to the duration of AHR signaling. I agree that this is the case. Yet, there are studies that show for some PAHs (e.g., benzo(a)pyrene), it is their metabolites that cause toxicity, not the parent compound. Mentioning this bolsters their overarching point about the complexity of AHR signaling, and the many unknowns that have yet to be resolved.

We have modified the text to take into account these comments.

2,3,7,8-tetrachlorodibenzo-p-dioxin is misspelled (it is missing 'para' (-p-) between dibenzo and dioxin).

This has been corrected.

Referee #2 (Remarks for Author):

The topic of the review article entitled "The influence of environmental signals via AHR on immune and tissue biology" is timely, very interesting and potentially important, in particular with regards to the potential targeting of AHR in clinical situations. However, **this article is not well structured and partially redundant, especially with regards to the authors' hypothesis that the majority of "so-called" AHR ligands act indirectly via inhibition of CYP1A1 and accumulation of endogenous AHR ligands.** In my eyes, the

authors do not pay sufficient attention to the growing list of studies investigating the different facets of AHR signaling as well as the AHR-dependent mode-of-action of certain chemicals. **In its current form, the article seems to reflect the very subjective opinion of the authors, while lacking an open and critical discussion of findings that do not necessarily align with this opinion.**

Points of concern:

1. Manuscript title is misleading. This article pays little to no attention to the impact of environmental signals on AHR.

We concede that the emphasis on environmental signals in the title may be misleading as of course not only endogenous environment plays a role but also the many exogenous factors that may influence AHR activity. As we did not want to make this review a toxicology vs physiology contest we think modification of the title is warranted.

2. Page 2, 2nd paragraph, l. 12: "The major metabolising enzyme which is highly induced by AHR activation is Cyp1a1,..."

This is probably true for some but not all tissues (liver > CYP1A2), cell-types, and conditions.

We have qualified this by referring to Cyp1a1 as the major Cyp1 family enzyme outside the liver. Cyp1b1 is not entirely controlled by AHR so we wanted to emphasise the role of Cyp1a1 in tissues other than the liver.

3. Page 2, 2nd paragraph, l. 14: "The detoxification aspect dominated the view on AHR over decades and restricted the interest to the fields of pharmacology and toxicology. This view has changed considerably over the last decade..."

Not true. Detoxification is not the same as toxicity. In general, this statement omits numerous important discoveries in the AHR field in various contexts, including immunology and cancer biology, that have been published way before 2014.

We agree that the term 'detoxification' was wrong here and have changed it. We also took out mention of the past decade as indeed a change in focus on AHR functions in the body began with the two Nature papers, Veldhoen et al and Quintana et al in 2008.

4. Page 3, l. 11 from top: "..., most determinations of ligands were conducted in vitro using highly sensitive luciferase constructs based on the consensus AHR response element (XRE). There are two problems with this approach. Firstly, the signal may be artificially amplified and not reflect activation under physiological conditions in vivo, and the in vitro approach is complicated by the fact that all tissue culture media contain tryptophan and therefore may include its derivative 6-formylindolo[3,2-b]carbazole (FICZ)(Rannug et al, 1987), a high affinity AHR ligand which could be the reason for a positive signal(Rannug & Fritsche, 2006; Veldhoen et al, 2009). Secondly, it has been shown that many putative AHR ligands are not ligands at all but rather inhibit the negative feedback of AHR activation via enzymes of the Cyp1 family. Cyp1 enzymes normally biotransform ligands to facilitate their excretion and thereby terminate AHR signaling(Wincent et al, 2012)."

I strongly recommend to attenuate these statements. For sure there is a long list of compounds that are called AHR ligands based on respective reporter gene data only. However, on the other hand one could easily list several dozens of AHR ligands that have been identified by means of radioactive ligand binding assays. Is this also true for CYP1A1 substrates as indicated by "many putative AHR ligands are not ligands..."? And if so, I assume it has been shown in vivo rather than in an artificial in vitro assay? Also, I think that the maximum CYP1A1 response that can be expected by accumulating FICZ is rather low in comparison to the dose-dependent induction by numerous of the "so-called" AHR ligands. In addition, I want to emphasize that the authors' idea that more or less all AHR ligands are metabolized by CYP1 enzymes is simply not true and I'm not only thinking of metabolically stable dioxins. The phase I reaction is thought to polarize lipophilic chemicals, accordingly, it is unlikely that CYP1 enzymes metabolize polar "AHR ligands", such as tryptophan metabolites (doi:10.1177/1178646923118250).

While it is indeed correct that ligand metabolism by Cyp1a1 has really only been thoroughly described for FICZ and ICZ we would put forward the argument that the phenotype of the constitutively active Cyp1a1 mouse directly proves that endogenous ligands are metabolized such that their depletion results in the phenocopy of an AHR null mouse. We have included a sentence on Trp-metabolites to highlight they are not substrates for CYP1 enzymes, including the relevant reference.

5. Page 3, l.10: "Physiologic ligands as opposed to man-made pollutants are readily biotransformed by AHR induced Cyp1 family members which catalyse their oxidative metabolism leading to their inactivation and excretion. In contrast, man-made pollutants are largely poor substrates for these enzymes and as a consequence circumvent the negative feedback of AHR activation by Cyp1, interfering with its physiological function."

This is again a very general statement. Please provide references for the CYP1A1 catalyzed metabolism of physiological ligands other than FICZ/ICZ. Also, some PAHs as well as man-made drugs are quite good substrates for CYP1 enzymes.

We have modified the text to make this section more nuanced.

6. Page 3, l.6 from bottom: "Despite numerous reports of deleterious functions of environmental pollutants that activate AHR, the difference underlying the mode of action of physiological ligands vs pollutants remains mechanistically unexplained, but it is likely related to dysregulated kinetics of AHR signaling."

This is again a very general statement and partially not true. What about the established Adverse Outcome Pathways involving AHR activation by environmental pollutants (see websites of OECD, US EPA, etc.)? In addition, the mechanisms by which polycyclic aromatic hydrocarbons are converted into highly genotoxic and/or redox-cycling metabolites are well-known. In addition, the authors should explain what they mean when stating "dysregulated kinetics of AHR signaling". In the context of xenobiotic metabolism and the definition of toxicokinetics and -dynamics, this is somehow confusing.

We have modified the sentence referring to dysregulated AHR signaling also in response to Reviewer 1 as indeed it remains unclear what exactly the underlying mechanisms of toxicity are. Adverse Outcomes are merely descriptors for negative events but have not so far provided any mechanistic insight. So far, there are only three WPHA/WNT endorsed AOPs in the AOP Wiki which link AOP activation with the adverse outcomes "increased early life stage mortality" and "uroporphyrinuria". As AOPs by design are chemical unspecific, these do however not add to unraveling the dichotomy between activation by a natural ligand causing no toxicity and a xenobiotic causing toxicity. Perhaps in the future, when quantitative AOPs are more developed, these will be useful for clarifying this question.

7. Page 3, l. 2 from bottom: "The anti-inflammatory consequences of AHR activation are well documented..."

Please specify. The TCDD-induced and AHR-mediated upregulation of COX2, for instance, is probably not an anti-inflammatory event. I assume this statement refers to physiological AHR signaling?

When we are referring to AHR activation we restrict this to physiological activation and of course we agree that whatever TCDD may do may well result in inflammatory action.

We have added a sentence however to make clear that not all AHR activities may be anti inflammatory (although we never actually said this, but rather mentioned 'predominant' AHR activities being anti inflammatory).

8. Pages 3-5: A few comments to the passage "Common AHR functions across tissues":

In the context of gut-derived microbial metabolites, it might be fair to mention that a lot of these compounds do not exclusively act via AHR but also via PXR (doi:10.1016/j.immuni.2014.06.014).

When discussing the role of AHR in endothelial cells and mentioning the vascular effects observed in AHR KO mice, is it of any relevance that those vascular phenotypes are not present in the AHR KO rat and thus might be species-specific? (doi:10.1016/j.taap.2013.06.024).

The Wiggins reference on endothelial cells also looked at human endothelial cells and found similar features to what they reported in mice.

9. Pages 6-7: The chapter "Mechanisms of AHR function" appears rather premature and does not deliver what it promises. In fact, it is more or less a continuation of the previous chapter, enumerating phenotypes of AHR knockout etc. A thorough and comprehensive overview about the manifold mechanisms of AHR signaling and associated functions is missing. The authors did not spend a single word on all the alternative pathways of AHR that have been identified in numerous studies to influence the outcome of AHR activation during immune and developmental processes as well as inflammatory responses. The authors instead focus on potential XRE and ADME parameters of the ligands, which for sure helps to explain some but certainly not all of the dichotomous effects evoked by AHR.

We have deliberately omitted discussion of the reported 'manifold alternative AHR signaling mechanisms as these are largely reported in cell lines and not mechanistically proven in vivo. We included a sentence now to refer to this and point to a published mouse model generated by Bradfield which could solve this issue in vivo but does not seem to have been used to address these postulated alternative AHR mechanisms of action.

10. Page 6, l. 15 from bottom: "Xenobiotic activation of the AHR at adult life stages causes less detrimental effects compared with the developmental stages,..."

Somehow weird to weigh detrimental health effects, especially when thinking in the human context. What about the central role of AHR in PAH exposure-associated cancers and other detrimental health consequences? Or do the authors only consider fish and mouse models when stating that?

Given the limitations in research with humans in vivo these statements indeed refer to mouse and fish models in which such events can be investigated.

11. Page 8, l. 110: "... it is conceivable that the therapeutic action of tapinarof is not due to direct AHR activation but rather inhibition of Cyp1a1 mediated metabolism of naturally present ligands such as FICZ or skin microbiota metabolites."

This is highly speculative and should be rephrased. Do the authors think that coal tar acts in a similar manner?

Coal tar contains PAHs which are known AHR ligands so no, we do not think it acts in a similar manner. The sentence has been slightly modified to meet the comment.

Minor corrections/comments:

12. Abstract, l. 3: Please write "2,3,7,8-Tetrachlorodibenzo-p-dioxin" instead of "2,3,7,8-Tetrachlorodibenzodioxin"

This has been corrected

13. Abstract, 3rd sentence: One or more words are missing "Notably AHR activity is important in preventing excessive inflammation following tissue damage in barrier organs such as skin, lung or gut has received wide attention in the past decade."

We thank the reviewer for pointing this out and have corrected the sentence.

14. Page 2, 2nd paragraph, l. 10: "It was identified because of its role in the induction of drug metabolizing enzymes leading to the detoxification of xenobiotics such as polycyclic aromatic hydrocarbons (Poland et al, 1976)."

Wrong reference, please correct. Poland et al. have discovered AHR as a protein that binds to TCDD, which is neither a PAH nor efficiently detoxified via AHR/CYP1.

We have rephrased the sentence to make clear what was shown in the Poland study.

Referee #3 (Remarks for Author):

This is an important review topic. Clarification of AHR function with a modern understanding of physiological vs. the traditional understanding derived from toxicological studies is needed. The story is complex, and thus requires clarity in writing. The ideas expressed in the review are important and timely. The writing can be improved. See specifics below:

Introduction:

"major molecular entry points for environmental factors is the aryl hydrocarbon receptor (AHR), a ligand dependent transcription factor belonging to the bHLH-PAS family of transcription factors which sense different aspects of the environment."

Does this mean that the entire family senses the environment, or is this a specific function of the AHR. Please clarify.

We have mentioned other members of the PAS family and their particular environmental sensing in the first paragraph of the introduction.

Main Body

"AHR expression on colonic neurons requires an intact microbiota which contrasts with intestinal epithelium which expresses AHR also in germfree mice."

I am not sure what this means. Please clarify

We have expanded this sentence to make the meaning clearer.

Mechanisms of AHR Function

I am not sure the first couple of paragraphs of this section should be characterized as mechanisms. For me mechanisms are more molecular-based, biochemical and molecular processes. This seems more overall inflammatory systems.

We have modified the heading of this section to clarify this point.

Fig.2 which shows the distribution of Cyp1a expression in zebrafish as a consequence of AHR activation by

either TCDD or FICZ.

This is an essential piece of the authors' argument. A more detailed explanation of the consequences of this kind of function is warranted.

We have extended the description in the text as well as figure legend to clarify.

"Altogether it is difficult at present to claim that there is a coherent picture of how AHR activation plays out in molecular terms."

I am not sure what this actually means.

We have modified this sentence.

AhR as a therapeutic target.

This is an important part of the review. However, the section seems to be a little chaotic and wandering. It needs better organization to understand the potential roles for AhR-based therapeutics across different disease states and cell types.

It is interesting that the reviewer thinks the section chaotic as in our opinion this could be said about the state of AHR driven therapy approaches. We have taken out some of the redundancy that was mentioned and added additional AHR targeting approaches under way in the cancer field. Given that there are numerous reports of AHR as a tumour suppressor as well as studies showing alleviating of clinical phenotypes via AHR ligands it is unavoidable that this section appears internally contradictory just as the field currently reflects this.

14th Aug 2024

Dear Dr. Stockinger,

Thank you for the submission of your manuscript to EMBO Molecular Medicine and please accept my apologies for the delay in getting back to you due to the holiday season. I am pleased to inform you that we will be able to accept your manuscript pending the following final amendments:

- 1) Please implement referee #2 suggestion.
- 2) Add up to 5 keywords.
- 3) Please define corresponding authors on the title page.
- 4) Please enter all funding information in the "Acknowledgments".
- 5) Rename "Competing interests statement" to "Disclosure and competing interests statement". We updated our journal's competing interests policy in January 2022 and request authors to consider both actual and perceived competing interests. Please review the policy <https://www.embopress.org/competing-interests> and update your competing interests if necessary.
- 6) Rename Figure 3 to Box 1.

I look forward to receiving the revised version of your manuscript.

Yours sincerely,

Zeljko Durdevic

*** IMPORTANT INFORMATION ***

- 1) a .doc formatted version of the manuscript text (including Figure legends and tables)
- 2) Separate figure files
- 3) a letter INCLUDING the reviewer's reports and your detailed responses to their comments.

Also, and to save some time should your paper be accepted, please read below for additional information regarding some features of our research articles:

- 1) Glossary: EMBO Molecular Medicine articles will be accompanied by a glossary explaining some of the terms used for laymen. I identified the following:

_____, _____, _____

Could you please help us in identifying terms that may need an "explanation" other terms that we can add to the glossary.

- 2) For more information: This is a short list of related web links for further consultation by the readers. Could you identify some relevant ones? Examples are patient associations, OMIM related links, databases, authors websites, etc.

- 3) Pending issues: At the end of each article we will have a box highlighting issues that still need further studies and where

research efforts should converge (we call this the Pending issues box). From my reading I would say:

but I can see there may be many more. Could you work on this as well?

4) Disclosure and competing interest statement: Please include a statement declaring any competing commercial interests in relation to your submitted work.

5) Please note that we now mandate that all corresponding authors list an ORCID digital identifier. This takes <90 seconds to complete. We encourage all authors to supply an ORCID identifier, which will be linked to their name for unambiguous name identification.

Currently, our records indicate that the ORCID for your account is 0000-0001-8781-336X.

Link Not Available

-

Thank you,

Zeljko Durdevic

***** Reviewer's comments *****

Referee #2 (Remarks for Author):

The revision of the review article has significantly improved its quality and stringency. With one exception, I'm satisfied with the response of the authors to my comments and recommend in principle that the paper be accepted for publication in EMBO Molecular Medicine. However, I would ask the authors to delete the term "the assumption" in the sentence "A further complication is the assumption that AHR may have non-genomic functions (reviewed in (Bock, 2020))." on page 7 of the revised MS. This expression is provocative and unnecessary. I think the authors' doubts with regard to the non-canonical functions become more than clear in the following sentence.

Referee #3 (Remarks for Author):

The authros have satisfied all the previous critiques.

The authors addressed the remaining editorial issues.

19th Aug 2024

Dear Dr. Stockinger,

We are pleased to inform you that your manuscript is accepted for publication and is now being sent to our publisher to be included in the next available issue of EMBO Molecular Medicine.

You will soon be contacted by Springer Nature to sign your publishing license. When you login to the customer service website, please use the token/code copied below to waive the article publication charges. Should you experience any difficulty, please email publishing@embo.org.

Waiver token: XXXXXXXXXXXXXXXXX

Your manuscript will be processed for publication by EMBO Press. It will be copy edited and you will receive page proofs prior to publication. Please note that you will be contacted by Springer Nature Author Services to complete licensing information.
